# The long-term effect of short point of care ultrasound course on physicians' daily practice

Ortal Tuvali[1], Re'em Sadeh[1,2]*, Sergio Kobal[1,3], Shaked Yarza[1,2], Yael Golan[4], Lior Fuchs[1,5]

**1** Joyce and Irwing Goldman Medical School, Ben-Gurion University of the Negev, Beer-Sheva, Israel, **2** Clinical Research Center, Soroka University Medical Center and The Faculty of Health Sciences, Ben-Gurion University of the Negev, Beer-Sheva, Israel, **3** Cardiology Department, Soroka University Medical Center and The Faculty of Health Sciences, Ben-Gurion University of the Negev, Beer-Sheva, Israel, **4** Internal Medicine F, Soroka University Medical Center and The Faculty of Health Sciences, Ben-Gurion University of the Negev, Beer-Sheva, Israel, **5** Emergency Department, Soroka University Medical Center and The Faculty of Health Sciences, Ben-Gurion University of the Negev, Beer-Sheva, Israel

☯ These authors contributed equally to this work.
* reem.sadeh90@gmail.com

**Data Availability Statement:** All relevant data are within the manuscript and its Supporting Information files.

**Funding:** The author(s) received no specific funding for this work.

## Abstract

### Background

The benefits of Point of Care Ultrasound (POCUS) are well established in the literature. As it is an operator-dependent modality, the operator is required to be skilled in obtaining and interpreting images. Physicians who are not trained in POCUS attend courses to acquire the basics in this field. The effectiveness of such short POCUS courses on daily POCUS utilization is unknown. We sought to measure the change in POCUS utilization after practicing physicians attended short POCUS courses.

### Methods

A 13-statements questionnaire was sent to physicians who attended POCUS courses conducted at the Soroka University Medical Center between the years 2014–2018. Our primary objective was to compare pre-course and post-course POCUS utilization. Secondary objectives included understanding the course graduates' perceived effect of POCUS on diagnosis, the frequency of ultrasound utilization and time to effective therapy.

### Results

212 residents and specialists received the questionnaire between 2014–2018; 116 responded (response rate of 54.7%). 72 (62.1%) participants were male, 64 (55.2%) were residents, 49 (42.3%) were specialists, 3 (2.5%) participants did not state their career status. 90 (77.6%) participants declared moderate use or multiple ultrasound use six months to four years from the POCUS course, compared to a rate of 'no use at all' and 'minimal use of 84.9% before the course. 98 participants [84.4% CI 77.8%, 91.0%] agree and strongly agree that a short POCUS course may improve diagnostic skills and 76.7% [CI 69.0%,

**Competing interests:** The authors have declared that no competing interests exist.

84.3%] agree and strongly agree that the POCUS course may shorten time to diagnosis and reduce morbidity.

## Conclusions

Our short POCUS course significantly increases bedside ultrasound utilization by physicians from different fields even 4 years from course completion. Course graduates strongly agreed that incorporating POCUS into their clinical practice improves patient care. Such courses should be offered to residents and senior physicians to close the existing gap in POCUS knowledge among practicing physicians.

## Introduction

Point-of-Care Ultrasound (POCUS) is being utilized for diagnostic and therapeutic purposes worldwide in many medical fields [1]. Its implementation as part of the physical examination improves patient management and reduces diagnostic uncertainty and improving accuracy [2–4].

The advantages of POCUS are well established in the literature; its vast assistance in the assessment and diagnosis of patients presenting with hypotension, chest pain or acute dyspnea [5–7], as well as, the diagnosis of DVT and guiding invasive procedures such as tracheostomy, central venous and arterial line insertion, paracentesis and thoracentesis [8, 9].

Nevertheless, POCUS is an operator-dependent modality which requires operators to be familiar with the technique in order to successfully obtain images and interpret them correctly [10]. Obtaining ultrasound images is especially difficult when it comes to obese and emphysematous patients, due to poor penetration of the sonographic waves; such cases require a highly skilled operator [11]. Due to the COVID-19 pandemic, the use of CT and CXR is restricted, thus making POCUS an ever more valuable tool. Considering the above, proper training in POCUS is necessary [12].

In recent years, POCUS training has been integrated into many medical schools' curriculum [13]. Physicians who were not trained in POCUS in medical school attend different short POCUS courses, hoping to acquire the basics in this field. The effectiveness of such short POCUS courses, on real life daily POCUS utilization is still unknown.

Only few studies have measured the impact of POCUS courses on physicians' ultrasound performance and interpretation abilities [14–16]. None described changes in ultrasound utilization in physicians' daily practice pre and post short POCUS course, months to years after such courses were attended by physicians.

We sought to measure the change in POCUS daily utilization, post short POCUS courses, among practicing physicians.

## Methods

A five-day POCUS course is conducted once to twice a year at the Soroka University Medical Center since 2013. The course is designated for physicians from different specialties, such as anesthesia, critical care, internal medicine, emergency medicine, and pediatrics. The course is designed for physicians with no previous POCUS training.

Course participants undergo five days of training, including over 13 hours of hands-on practice and 18 hours of lectures. In short, the course syllabus includes cardiac ultrasound anatomy, POCUS utilization for the differential diagnoses of shock, POCUS for vascular

access, lung ultrasound, assessment of volume status and fluid responsiveness as well as exposure to numerous relevant clinical cases (S1 Appendix: Course curriculum).

A 13-statements questionnaire was sent to all the physicians that participated in the courses which took place between the years 2014–2018 (S2 Appendix: Questionnaire). The questionnaire was sent to all participants at the same time. The time gap from course graduation to the filling of the questionnaire varied between responders but was not less than six months and up to four years from course graduation for all participants. This time gap was chosen to assess the true long-term effect of the short POCUS course on reported ultrasound utilization and physicians' perception of this modality. Participants were asked to select their level of agreement with the questionnaire's various statements, using the responses: "Strongly disagree"; "Disagree"; "Neutral"; "Agree", and "Strongly agree", scaled from 0–4.

The questionnaire was designed primarily to compare pre-course and post-course POCUS utilization. Secondary objectives were to describe course graduates' perceived effect of POCUS on the diagnostic abilities, the frequency of ultrasound utilization, the time to effective therapy, on patients' outcomes, and on the level of confidence to use this bedside modality (S2 Appendix: Questionnaire). Baseline data as a type of physicians' specialty and seniority was also collected.

The Soroka University Medical Center ethical committee granted waver for informed consent and approved the research.

## Statistical analysis

Population characteristics were shown with numbers and percentages.

Trend test and Chi square test were used to analyze the pre and post course answers about the frequency of use of POCUS. This analysis was assessed in the general population, and in subgroups of gender, degree, year of course, and location of the hospital. In addition, the delta between the pre and post course answers, divided by years, was shown in a bar chart.

The answers for other questions were displayed in a bar chart and a distribution table that show how many people answered each answer and the percentages received for each answer. Reliability analysis was conducted on the questionnaire, and Cronbach's alpha was calculated to show the reliability accepted for each questionnaire ($\alpha = 0.84$). All the questions would show no significant increase in the alpha if they were deleted.

All statistical tests were performed at $\alpha = 0.05$ (2-sided) using IBM SPSS software, version 25.

## Results

212 residents and specialists received, between 2014–2018, a web-based questionnaire at least six months and up to 4 years following completion of the course they attended. Of them, 116 responded (response rate of 54.7%). 72 (62.1%) participants were male, 64 (55.2%) were residents, 49 (42.3%) were specialists, 3 (2.5%) participants did not state their career status. Participants' medical fields and additional baseline characteristics are shown in Table 1.

## Ultrasound utilization increased significantly after the course

Ultrasound utilization post POCUS course increased significantly with a rate of 'no use at all' and 'minimal use only' of 84.9% before the course, which dropped to 22.4% after the course. 90 participants (77.6%) declared "moderate use" "or multiple use" post POCUS course compared to 18 participants (16%) before the course (P- value < 0.001, P for Trend = 0.01) (Fig 1).

**Table 1. Participants characteristics.**

| Variables | | Number of participants (n %) |
|---|---|---|
| Gender, N males (%) | | 72 (62.1%) |
| Degree, N residents (%) | | 64 (55.2%) |
| Specialty, N (%) | Anesthesia & Critical Care | 27 (23.3%) |
| | Cardiology & Cardiac Surgery | 3 (2.6%) |
| | Emergency Medicine | 10 (8.6%) |
| | Internal Medicine | 65 (56%) |
| | Surgery | 5 (4.3%) |
| | Pediatric | 4 (3.4%) |

## The perceived effect of POCUS course on patients' care

90 participants [75.8%, CI 68.01%,83.59%] agree or strongly agree with the statement that incorporating POCUS into their practice will improve the care they provide to patients (Fig 2).

## The perceived effect of POCUS course on the quality of diagnosis

98 participants [84.4%, CI 77.8%, 91.0%] agree and strongly agree with the statement that a short Point of Care ultrasound course may improve diagnostic skills (Fig 3).

Fig 4 illustrates that most participants agree and strongly agree [92.2% CI 87.32%,97.88%] with the statement that point of care ultrasound is a diagnostic modality that can provide a more accurate and quicker diagnosis for several disease processes.

In Fig 5, 76.7% [CI 69.0%, 84.3%] of participants agree and strongly agree that the POCUS course may shorten time to diagnosis, thus may reduce morbidity.

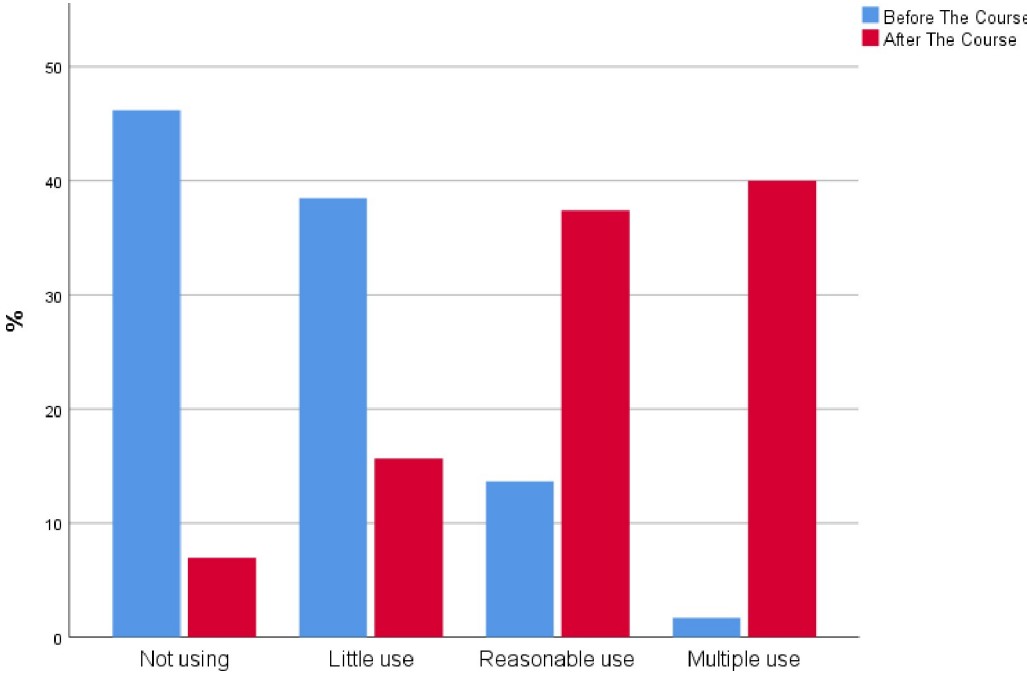

**Fig 1. Ultrasound utilization in daily practice.**

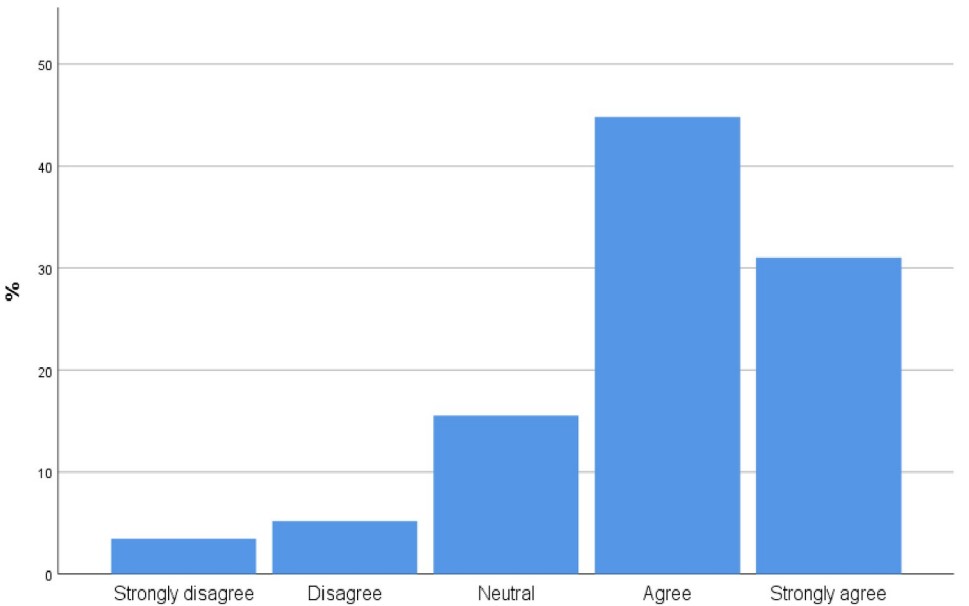

**Fig 2. To what extent do you believe incorporating POCUS into your practice will improve the care you provide patients?**

The results in Fig 6 from the statement: "Incorporation of POCUS as part of physicians' daily practice will influence patients' outcome and recovery" showed that 31.9% [CI 23.4%, 40.3%] participants agree or strongly agree with the statement while about half of participants (44.8% [CI 38.76%, 53.84%]) moderately agreed with the statement above.

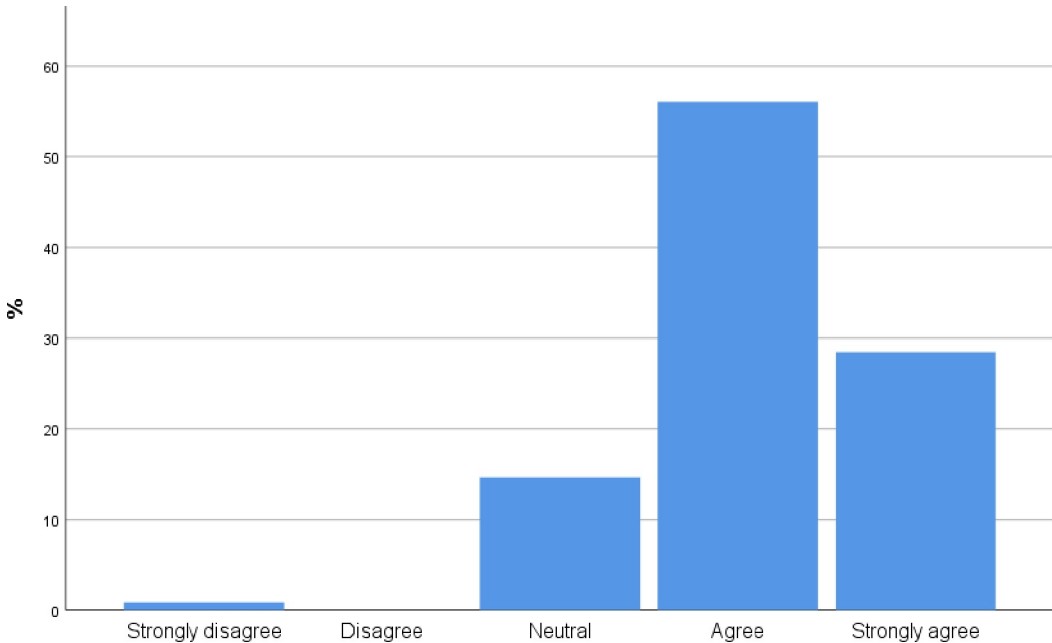

**Fig 3. To what extent do you agree with the following statement: A short point of care ultrasound course improve diagnostic skills?**

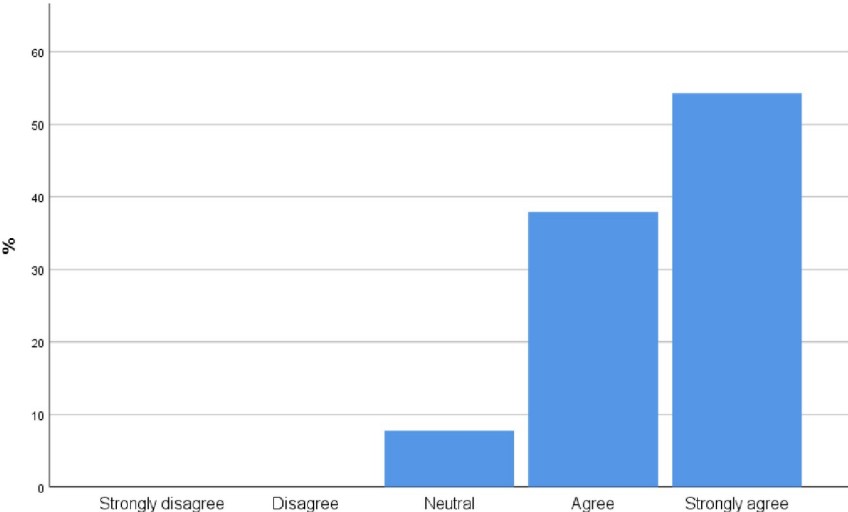

**Fig 4. To what extent do you agree with the following statement: Point of care ultrasound is a diagnostic modality that can be used by clinicians to provide more accurate and quicker for several disease processes?**

The great majority of responders (110 participants, 94.8% [CI 86.4%, 100%]) agree and strongly agree that the POCUS course should be an integral part of both residents' and specialists' medical training (Fig 7). In addition, as Fig 8 shows, most POCUS course graduates (85.3% [CI 78.86%, 91.74%, (P value = 0.04)] agree and strongly agree that their ultrasound skills improved following the course.

Only 24 (20.7%) have had some previous training in this field (Fig 9). Furthermore, 87% participants would recommend their colleagues to take part in the POCUS course (Fig 10).

Of 116 participants, 66 (56.8%) graduates stated a great or extreme intention to integrate POCUS into their routine clinical care after the course (Fig 11). This is while there was no association between the time that elapsed from the course (from six months to four years) and the delta in ultrasound utilization pre and post the course (Fig 12).

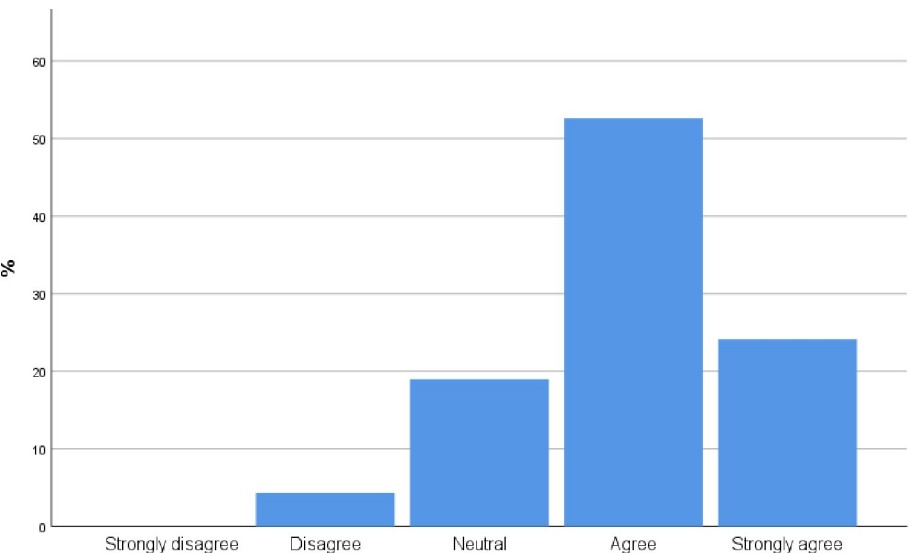

**Fig 5. Do you agree with the following statement: Short POCUS course may shorten time to diagnosis thus may reduce morbidity?**

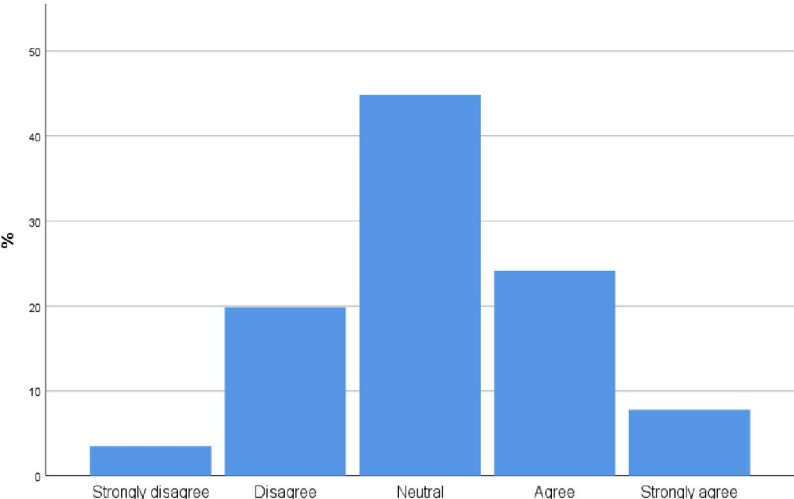

**Fig 6. Do you think incorporating POCUS as a part of your daily practice may influence on your patients' outcome and recovery (e.g. length of hospital stays, admission rate etc.)?**

As for the limitations of POCUS technique, 78.4% of participants proclaimed a great or extreme comfort with the ultrasound machine and their ability to operate it within different clinical scenarios after the course (Fig 13).

Among all participants, 86.2% [CI 79.93%, 92.47%] agree and strongly agree with the statement that they feel comfortable with their understanding of the capabilities and limitations of POCUS (Fig 14).

## Discussion

In this study, we found that a short POCUS course has a great impact on the daily practice of physicians, increasing ultrasound utilization, improving perceived patients care, and

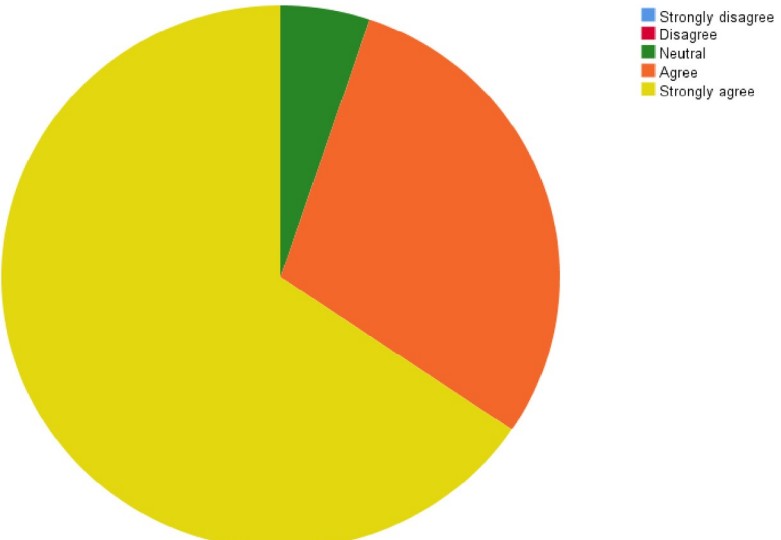

**Fig 7. Do you think POCUS course should be an integral part both residents' and specialists' medical training?**

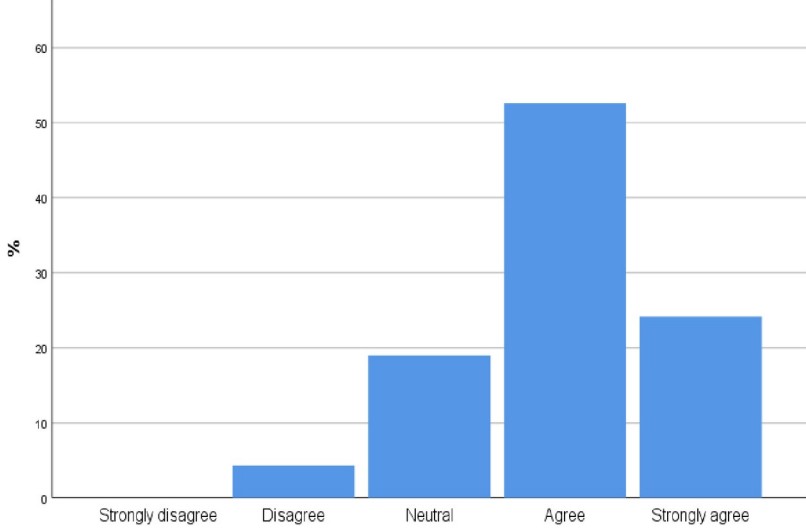

**Fig 8. Have your ultrasound skills improved due to this course?**

improving perceived physicians' diagnostic skills. This was true even four years from the course and across all subspecialties.

To the best of our knowledge, this is the first study to show that teaching practicing physicians how to conduct and utilize bedside ultrasound in various clinical scenarios and, in a relatively short period time, changes physicians' daily practice by increasing the frequency of ultrasound utilization at the bedside. Moreover, participants reported that they recognize the importance and contribution of integrating POCUS as an integral part of their daily practice. They also see it as a tool that improves diagnostic skills and enables a faster and more accurate diagnosis.

Only few studies describe different POCUS course methods and their impact. A previous prospective study of 17 Rwandan physicians participating in a 10-day-long POCUS training, in a resource-limited setting, found an improvement in the participants' knowledge, job satisfaction, and patient care. They did not measure the change in daily ultrasound utilization.

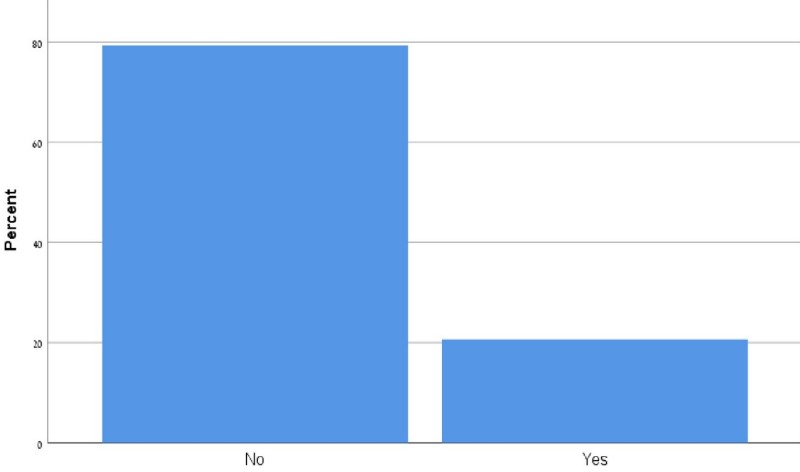

**Fig 9. Have you ever had similar previous training in this filed?**

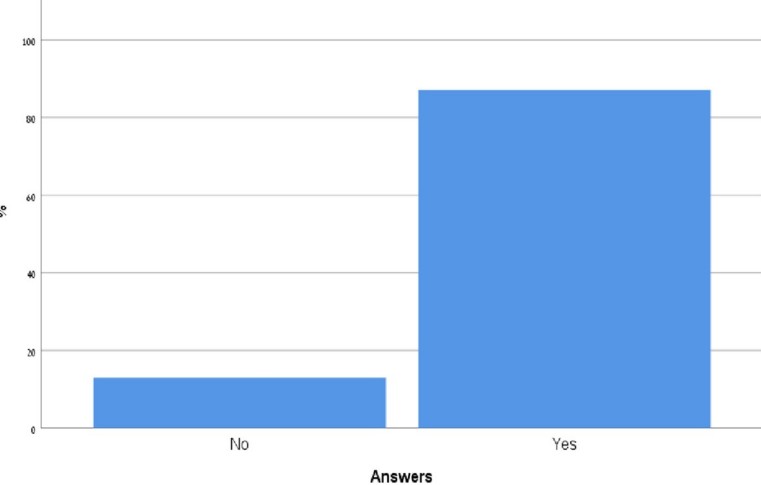

**Fig 10. Would you recommend this course to your colleagues?**

Neither did they measure its utilization one year or more after the course (14). Another study examined the impact of a course of seven sessions, each session two hours long. 16 surgical residents participated in the study, in which it was found that surgical residents showed an improvement in self-efficacy and confidence levels across a broad range of skills [15].

An additional study measured the impact of a single-day intensive bedside ultrasound workshop followed by two optional hour-long workshops for 33 internal medicine interns, revealing a significant increase in assessment performance [16].

Our study shows that a relatively short, 5-day-long POCUS course is successful in changing both residents' and specialists' daily routine, as long as four years after the course, among a

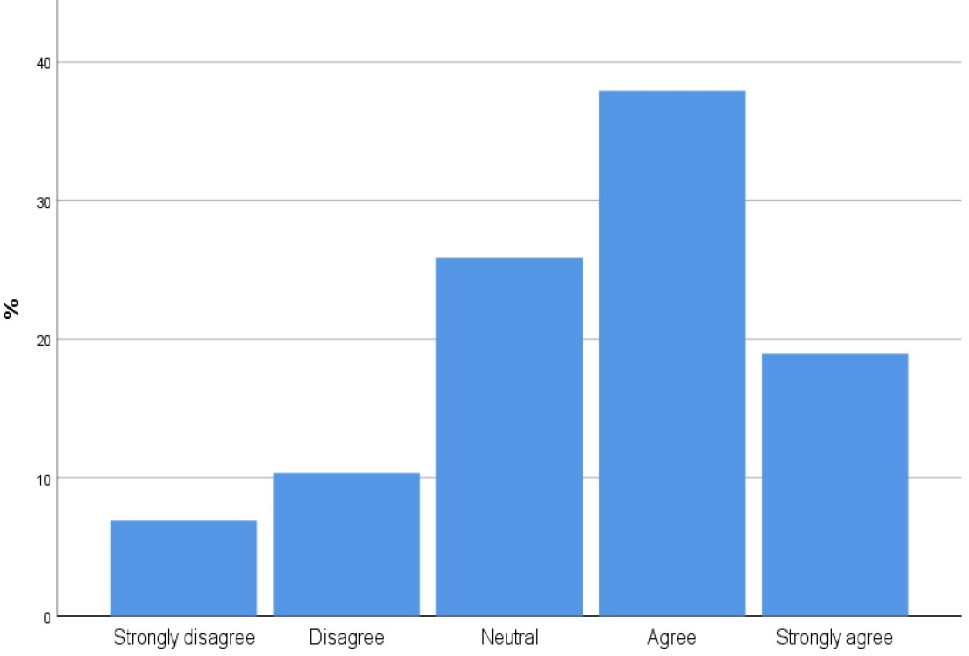

**Fig 11. How likely are you to integrate POCUS into your clinical care after the course?**

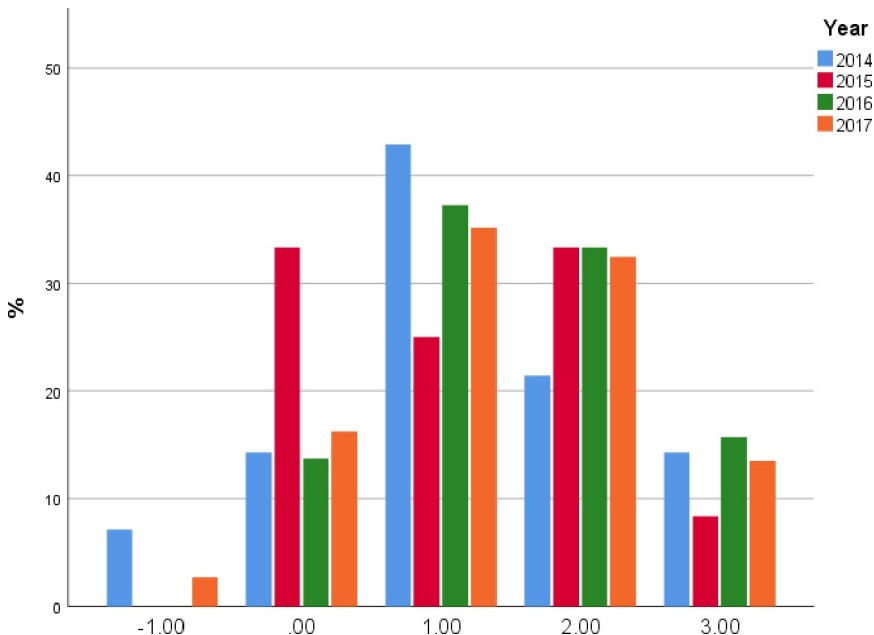

**Fig 12. Delta of the frequency of US utilization before and after the course, stratified by year; there was no association found between the delta of US utilization and the time that passed since training (S3 Appendix).**

large number of participants from many different subspecialties. We bring forward evidence that it is feasible to teach novice physicians, in a short POCUS training, how to conduct bedside lung and cardiac ultrasound and implementing it in their daily practice.

We found that course graduates strongly agree with the perception that this modality is significantly helpful in the diagnosis and management of their patients, eventually assisting the physicians in performing more productively. This perception explains why there is an increase

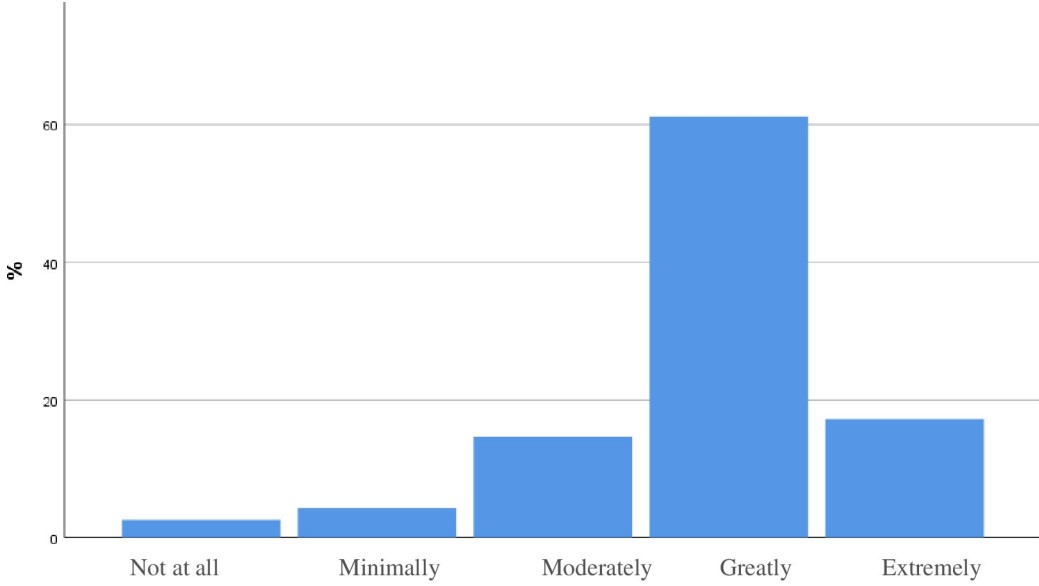

**Fig 13. How comfortable do you feel with your understanding of the ultrasound machine and your ability to operate it within different clinical scenarios after the course?**

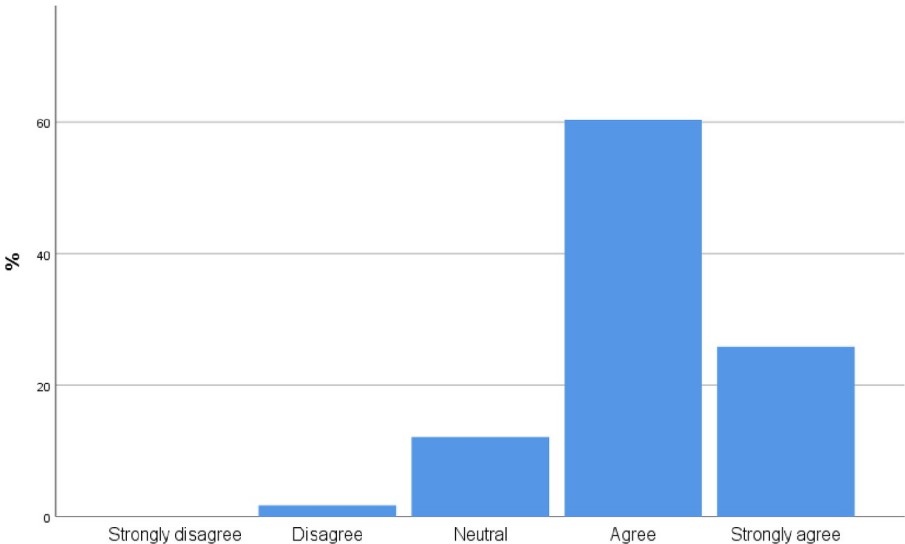

**Fig 14. How comfortable do you feel with your understanding of the capabilities ad limitations of POCUS?**

in the utilization of POCUS in their daily practice. However, surprisingly enough only 31.9% agreed that POCUS may influence their patients' outcome and recovery, such as shortening hospitalization duration, reducing admission rate, etc. In addition, we showed that a short course could improve physicians' bedside skills and change patient management in the long term. We found no association between time from the course and the frequency of ultrasound utilization (S3 Appendix). Even years after the course, the increased ultrasound daily utilization continued, showing that this is a real practice change and not only short-term course effect. For our course graduates, this course initiated a learning process that resulted in a change in clinical practice increasing their ultrasound utilization significantly and altering their patients' bedside assessment and care.

There are several limitations to this study. The primary objective was answered by comparing pre and post POCUS course data, derived from the questionnaire, showing a significant increase in daily POCUS utilization. Even though the course participants demonstrated a significant change in patient management as well as potential to reduce morbidity and mortality (Figs 2–13), increasing their confidence in their daily work, this does not guarantee that they are fully competent in the use of ultrasound. Further research is needed in order to understand the direct effect of a short POCUS training process on patient management and on physicians' POCUS skills.

Second, like any questionnaire-based study, there is an integrated selection bias stemming from the decision of some of the course participants to answer the questionnaire and some who chose not to. Nevertheless, the relatively high response rate of 55% may reduce the effect of this bias.

Lastly, all the presented data is subjective. We did not perform any objective measurements of ultrasound utilization before or after the course (such as the number of used gel bottles). However, all participants served as their own historical controls, comparing each participant's ultrasound utilization rate prior to the course to the utilization at the time of the questionnaire.

There is a knowledge gap between physicians trained in POCUS during medical school and residency and those who were not. In this era of COVID-19, POCUS was utilized repeatedly at the bedside, as other imaging modalities were less available. This illustrates the importance of

this modality. Nowadays, we face a challenge of how to increase the know-how of this effective, augmented, physical bedside examination [17].

## Conclusions

Our five-day POCUS course significantly increases bedside ultrasound utilization by physicians from different fields in the short and long term, as long as four years from graduation. Course graduates strongly agreed that incorporating POCUS into their clinical practice improves the care they provide to patients. Such courses should be offered to residents and senior physicians to close the existing gap in POCUS knowledge among practicing physicians.

## Supporting information

**S1 Dataset. Participants answers.**
(XLSX)

**S1 Appendix. Point of care ultrasound curriculum.**
(DOCX)

**S2 Appendix. 'Point of care ultrasound' course- questionnaire.**
(DOCX)

**S3 Appendix. Comparison of participants' answers stratified by year of the US course.**
(DOCX)

## Author Contributions

**Conceptualization:** Re'em Sadeh, Shaked Yarza, Lior Fuchs.

**Data curation:** Ortal Tuvali, Re'em Sadeh, Shaked Yarza, Yael Golan.

**Formal analysis:** Shaked Yarza.

**Investigation:** Sergio Kobal.

**Methodology:** Ortal Tuvali, Re'em Sadeh, Sergio Kobal.

**Project administration:** Sergio Kobal, Lior Fuchs.

**Supervision:** Yael Golan, Lior Fuchs.

**Visualization:** Ortal Tuvali, Re'em Sadeh, Shaked Yarza, Lior Fuchs.

**Writing – original draft:** Ortal Tuvali.

**Writing – review & editing:** Ortal Tuvali, Re'em Sadeh, Sergio Kobal, Yael Golan, Lior Fuchs.

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
