## [Decision Letter · Decision Letter 0]

1 Oct 2020

PONE-D-20-19295

The Long-Term Effect of Short Point of Care Ultrasound Course on Physicians’ Daily Practice

PLOS ONE

Dear Dr. Sadeh,

Thank you for submitting your manuscript to PLOS ONE. After careful consideration, we feel that it has merit but does not fully meet PLOS ONE’s publication criteria as it currently stands. Therefore, we invite you to submit a revised version of the manuscript that addresses the points raised during the review process.

We look forward to receiving your revised manuscript.

Kind regards,

Etsuro Ito

Academic Editor

PLOS ONE

Journal Requirements:

Additional Editor Comments:

The reviewers think your manuscript is worth for publication. But before that please revise your manuscript slightly.

Journal Requirements:

Reviewers' comments:

Reviewer's Responses to Questions

**Comments to the Author**

1. Is the manuscript technically sound, and do the data support the conclusions?

Reviewer #1: Yes

Reviewer #2: Partly

2. Has the statistical analysis been performed appropriately and rigorously? 

Reviewer #1: Yes

Reviewer #2: I Don't Know

3. Have the authors made all data underlying the findings in their manuscript fully available?

Reviewer #1: Yes

Reviewer #2: Yes

4. Is the manuscript presented in an intelligible fashion and written in standard English?

Reviewer #1: Yes

Reviewer #2: Yes

5. Review Comments to the Author

Reviewer #1: This manuscript primary's goal is to evaluate the incorporation of point-of-care ultrasound into the routine of physicians after a course given at Soroka University. The authors used a questionnaire to assess the primary outcome and other variables; each physician's pre-course time was used as control. The rate of responders was reasonable (55%), and overall, they felt the course had changed their practice.

Comments and suggestions to the authors:

Major comment:

From 116 responders(specified in the abstract and the results), 64 were residents, and 49 were specialists. What about the other 3? It is not specified throughout the manuscript.

Minor comments:

Figure 9: in the main text (pg 10, line 220), it is described that this figure is about previous training in this field. However, in the legend of the figure, it is not specified that it is previous training - consider specifying it.

Figure 12: it is not clear by the legend of the graphic's meaning. I realized it is the delta of changing in the answers, in which questions were categorized and attributed punctuation of 1. Although this information is in the main text, consider clarifying it.

Page 28: table "comparison of the answers to questions between the different years of the course" is without legend and enumeration. Also, this table is in the main text, readers are not aware of the questions' content. Consider moving this table to the appendix, as each question is described there.

Reviewer #2: Thankyou for the opportunity to review this manuscript.

This paper provides some evidence that attending an ultrasound course increases your use of ultrasound after. It also increase your confidence afterwards. Both these results are perhaps expected but may well not have been published before. My major concern is that you do not address the difference between the confidence of the respondents (which you present here) and their competence - the two are very different. And without assessing their competence we cannot know if patient care is improved, or morbidity reduced etc. We know that confidence and competence are not that well linked in educational terms (candidates who are confident and use much more ultrasound may not actually be competent to do so). This has to be acknowledged in the discussion in my view. The other things to be considered:

- I do not believe that the results section is the place to write that some results are controversial (line 209). Results should merely be stated in this section. The place to discuss their relevance (controversial or not) is the discussion. I would suggest this result is surprising rather than controversial

- I would be wary of stating that your response rate makes your results reliable. they are still subject to the extreme biases you mention (those that reply are by definition more likely to be interested in ultrasound and it's use). I agree the response rate is good, but I think a better phrasing would be that it reduces the chances of these biases' is a more appropriate sentence

6. PLOS authors have the option to publish the peer review history of their article (what does this mean?). If published, this will include your full peer review and any attached files.

Reviewer #1: No

Reviewer #2: No

---

## [Author Response · Author response to Decision Letter 0]

25 Oct 2020

Response to Reviewers

The long-term effect of short point of care ultrasound course on physicians’ daily practice

Dear reviewers,

We thank you very much for your comments. In light of the great impact that the POCUS course described in our paper has had on many physicians, we believe it is important to research this topic. We feel that these remarks have helped us to fine tune our paper and better understand its results.

Reviewer #1:

1. From 116 responders (specified in the abstract and the results), 64 were residents, and 49 were specialists. What about the other 3? It is not specified throughout the manuscript.

a. We have added this specification: “64 (55.2%) were residents, 49 (42.3%) were specialists, 3 (2.5%) participants did not state their career status”

2. Figure 9: in the main text (pg 10, line 220), it is described that this figure is about previous training in this field. However, in the legend of the figure, it is not specified that it is previous training - consider specifying it.

a. Completed

3. Figure 12: it is not clear by the legend of the graphic's meaning. I realized it is the delta of changing in the answers, in which questions were categorized and attributed punctuation of 1. Although this information is in the main text, consider clarifying it.

a. The legend was rephrased: “Delta of the frequency of US utilization before and after the course, stratified by year; there was no association found between the delta of US utilization and the time that passed since training (Appendix 3)”

4. Page 28: table "comparison of the answers to questions between the different years of the course" is without legend and enumeration. Also, this table is in the main text, readers are not aware of the questions' content. Consider moving this table to the appendix, as each question is described there.

a. We fully agree with this remark, changes were made

Reviewer #2

Thank you for the opportunity to review this manuscript.

This paper provides some evidence that attending an ultrasound course increases your use of ultrasound after. It also increases your confidence afterwards. Both these results are perhaps expected but may well not have been published before.

1. My major concern is that you do not address the difference between the confidence of the respondents (which you present here) and their competence - the two are very different. And without assessing their competence we cannot know if patient care is improved, or morbidity reduced etc. We know that confidence and competence are not that well linked in educational terms (candidates who are confident and use much more ultrasound may not actually be competent to do so). This has to be acknowledged in the discussion in my view

a. We agree, the following sentence was rephrased in the discussion: “Even though the course participants demonstrated a significant change in patient management as well as potential to reduce morbidity and mortality (fig 2-13), increasing their confidence in their daily work, this does not guarantee that they are fully competent in the use of ultrasound. Further research is needed in order to understand the direct effect of a short POCUS training process on patient management and on physicians’ POCUS skills.”

2. I do not believe that the results section is the place to write that some results are controversial (line 209). Results should merely be stated in this section. The place to discuss their relevance (controversial or not) is the discussion. I would suggest this result is surprising rather than controversial

a. We have rephrased controversial to surprising and moved it to discussion:“However, surprisingly enough only 31.9% agreed that POCUS may influence their patients’ outcome and recovery, such as shortening hospitalization duration, reducing admission rate, etc.”

3. I would be wary of stating that your response rate makes your results reliable. they are still subject to the extreme biases you mention (those that reply are by definition more likely to be interested in ultrasound and it's use). I agree the response rate is good, but I think a better phrasing would be that it reduces the chances of these biases' is a more appropriate sentence

a. Agree, the following sentence was phrased again: “Nevertheless, the relatively high response rate of 55% may reduce the effect of this bias”

---

## [Editor Report · Decision Letter 1]

27 Oct 2020

The Long-Term Effect of Short Point of Care Ultrasound Course on Physicians’ Daily Practice

PONE-D-20-19295R1

Dear Dr. Sadeh,

We’re pleased to inform you that your manuscript has been judged scientifically suitable for publication and will be formally accepted for publication once it meets all outstanding technical requirements.

Kind regards,

Etsuro Ito

Academic Editor

PLOS ONE

Additional Editor Comments:

Thank you for your revision.

---

## [Editor Report · Acceptance letter]

4 Nov 2020

PONE-D-20-19295R1 

The long-term effect of short point of care ultrasound course on physicians’ daily practice 

Dear Dr. Sadeh:

I'm pleased to inform you that your manuscript has been deemed suitable for publication in PLOS ONE. Congratulations! Your manuscript is now with our production department. 

Kind regards, 

on behalf of

Prof. Etsuro Ito 

Academic Editor

PLOS ONE